# Scalable Preparation of Complete Stereo-Complexation Polylactic Acid Fiber and Its Hydrolysis Resistance

**DOI:** 10.3390/molecules27217654

**Published:** 2022-11-07

**Authors:** Mingtao Sun, Siyao Lu, Pengfei Zhao, Zhongyao Feng, Muhuo Yu, Keqing Han

**Affiliations:** 1College of Materials Science and Engineering, Donghua University, Shanghai 201620, China; 2State Key Laboratory for Modification of Chemical Fiber and Polymer Materials, Donghua University, Shanghai 201620, China; 3HI-Tech Fiber Group Corporation, Beijing 100020, China; 4Shanghai Different Chemical Fiber Co., Ltd., Shanghai 201502, China

**Keywords:** PLA fiber, stereo-complexation, tension heat-setting, heat resistance, hydrolysis resistance

## Abstract

Due to their high sensitivity to temperature and humidity, the applications of polylactic acid (PLA) products are limited. The stereo-complexation (SC) formed by poly(L-lactic acid) (PLLA) and its enantiomer poly(D-lactic acid) (PDLA) can effectively improve the heat resistance and hydrolysis resistance of PLA products. In this work, the blended melt-spinning process of PLLA/PDLA was carried out using a polyester fiber production line to obtain PLA fiber with a complete SC structure. The effects of high-temperature tension heat-setting on the crystalline structure, thermal properties, mechanical properties, and hydrolysis resistance were discussed. The results indicated that when the tension heat-setting temperature reached 190 °C, the fiber achieved an almost complete SC structure, and its melting point was 222.5 °C. An accelerated hydrolysis experiment in a 95 °C water bath proved that the SC crystallites had better hydrolysis resistance than homocrystallization (HC). The monofilament strength retention rate of SC−190 fiber reached as high as 78.5% after hydrolysis for 24 h, which was significantly improved compared with PLLA/PDLA drawn fiber.

## 1. Introduction

With the gradual depletion of oil resources, the need to find new carbon sources becomes urgent and important [1]. In addition, environmental issues such as climate warming and white pollution caused by the large-scale use of petroleum-based derivatives have made it challenging to continue to use traditional petroleum-based polymers [2,3]. Polylactic acid (PLA) is a biomass material whose raw material, lactic acid, can be fermented from natural biological materials such as corn and straw. And it can be degraded into CO_2_ and H_2_O in nature, so it is one of the most ideal biodegradable materials [4]. Moreover, due to its favorable mechanical properties and processability, PLA is considered to be the most promising alternative to traditional petroleum-based polymers. PLA also has good spinnability, and can be made into monofilament or multifilament with round cross-sections, special-shaped fiber, fine denier fiber, crimped or non-crimped fiber, bicomponent fiber, spun-bonded nonwoven fabric, melt-blown nonwoven fabric, etc. Therefore, PLA fiber is expected to be used in apparel, household products and decoration, non-woven fabric, and in the hygiene and medical fields, etc. However, the textile dyeing and printing process needs to be carried out in a high-temperature and high-humidity environment. PLA products have a low softening point (60–70 °C) [5], therefore easily deform under heating conditions. Moreover, PLA is prone to hydrolysis in a high-temperature and high-humidity environment, which greatly reduces the performance of PLA fiber products.

Since Ikada et al. [6] first reported stereo-complexation (SC) formed by Poly(L-lactic acid)/poly(D-lactic acid) (PLLA/PDLA) blends in melt or solution, researchers have shown great interest in the crystallization behavior, crystal structure, and properties of SC [7,8,9]. Compared with ordinary homocrystallization (HC) composed of enantiomerically neat PLLA or PDLA 10_3_-helix chains, the enantiomers PLLA and PDLA chains can be arranged tightly side by side in a 3_1_-helix conformation, resulting from the driving effect of the intermolecular hydrogen bond interaction [10,11]. Due to dense chain stacking and stronger intermolecular interactions, SC crystallization exhibited better physical properties, such as mechanical strength, hydrolytic stability, thermal stability, and heat resistance [8,9,12,13,14]. In particular, the melting point (T_m_) of SC crystallization is as high as 220–230 °C (about 50 °C higher than the T_m_ of HC), and the Vicat-softening temperature of PLA with almost complete SC structure can reach 200 °C [5,15]. Pan et al. [16] proved that the fiber-softening temperature of pure PLA fiber was 72 °C, while that of PLA fiber with complete SC structure was up to 133 °C. 

However, it proved difficult to extend to the existing industrial-scale production for the commonly used stereo-complex methods. Stereo-complex processes related to dissolution usually have high production costs and low efficiency, and require complex equipment to collect solvents, which may cause environmental problems due to solvent disposal [17]. To achieve complete SC crystallization in PLLA/PDLA blends, additives were often used to promote nucleation (that is, the formation of spiral PLLA/PDLA pairs) and crystal growth, such as nucleating agents [18,19,20], plasticizers [21], and compatibilizer [22,23,24]. However, the addition of these additives will bring processing and economic difficulties. The strong shear force induced by electrospinning at high voltage (10–20 kV) was also found to be conducive to the formation of SC crystallization [25,26,27]. In addition, low-temperature sintering [28,29], oscillation shear injection molding (OSIM) [5], the introduction of supercritical fluid [17], and other methods have been used to prepare PLLA/PDLA blends with complete SC. However, the above-mentioned stereo-complex processes were complicated, uneconomical, and had low short-term industrialization potential. Since temperature and flow field are the key factors for SC formation, some researchers have prepared PLA fiber with a complete SC structure by melt-spinning on a laboratory scale [16,30,31,32]. The large-scale production of PLA products with complete SC structure from commercial high-molecular-weight PLA has recently become an issue worthy of attention.

In this work, commercially available high-molecular-weight PLLA and PDLA were used as raw materials and melt-spun into PLLA/PDLA fiber using a polyester fiber production line. The content of SC crystallites of PLLA/PDLA fiber was adjusted by high-temperature tension heat-setting. Thereby, the scalable preparation of complete stereo-complexation PLA fiber with favorable heat and hydrolysis resistances was achieved through a simple process.

## 2. Results

### 2.1. Crystallization Performance

WAXD (wide-angle X-ray diffraction) patterns showing the crystal structure and orientation structure results for different PLLA/PDLA fiber samples are illustrated in Figure 1. As seen in Figure 1a, there was an obvious diffraction peak near 2θ ≈ 16.4°, which corresponded to the (200) crystal face of HC crystallites. With the increase in the tension heat-setting temperature, the diffraction peak of HC gradually weakened and disappeared. At the same time, the diffraction peaks near 2θ ≈ 11.9°, 2θ ≈ 20.6° and 2θ ≈ 23.8° appeared, which are assigned to the (100), (300), and (220) crystal faces of SC crystallites, respectively. Table 1 lists the crystallization parameters of the fiber samples calculated according to the Figure 1. SC−PLA fiber obtained after tension heat-setting exhibited significantly higher crystallinity (X_c_) and stereo-complex rate ($f_{\mathrm{SC}}$) than the drawn−fiber. The decrease in crystallinity of SC−170 compared to SC−160 is due to the fact that part of the HC crystallites started to melt at 170 °C, but there was not enough energy to recrystallize and transform into SC for molten HC, and, thus, an amorphous structure formed during the cooling process. In contrast, SC−180 and SC−190 were more susceptible to HC-to-SC transformation at higher temperatures, and thus both the crystallinity and stereo-complex rate were increased. It is clearly seen that the stereo-complex rate of PLLA/PDLA fiber samples gradually increased with the increase in the tension heat-setting temperature. When the tension heat-setting temperature reached 190 °C, the stereo-complex rate reached as high as 98.7%. This demonstrated that the transformation of HC to SC was promoted as the tension heat-setting temperature increased. The crystal grain size corresponding to the (100) crystal face in the fiber also showed a gradually increasing trend. This was mainly because the molecular chain of PLA had a higher chain diffusion efficiency at higher temperatures, which was more favorable for the formation of SC. Therefore, an increase in the tension heat-setting temperature was conducive to an increase in the grain size and the proportion of SC crystallites to varying degrees in the PLLA/PDLA fiber. As Figure 1b and the orientation factor ($f_{C}$) in Table 1 show, as the tension heat-setting temperature increased, the orientation degree of the crystal regions inside PLLA/PDLA fiber gradually decreased, especially when the temperature reached 190 °C, when the $f_{C}$ of the crystal regions declined to 78.7%. This may be due to the occurrence of de-orientation of the crystal regions at excessively high heat-setting temperatures.

Figure 2 shows a schematic diagram of the stereo-complexation process for PLLA/PDLA fiber. Amorphous regions, HC crystallites, and SC crystallites coexisted in drawn−fiber (see Figure 1a). During the drawing process, HC crystallites and SC crystallites formed simultaneously. Due to the low drawing temperature (62 ± 2 °C), HC crystallites usually outnumber SC crystallites, even though SC is more thermodynamic favorable. This is because the enantiomers PLLA and PDLA must participate in a side-by-side manner in the stereo-complex process, and they will suffer from extended chain diffusion paths [7], resulting in a dominant ratio of HC crystallites in the crystalline region of drawn−fiber. Tension heat-setting treatment facilitated further crystallization of PLLA/PDLA fiber, leading to the formation of SC−PLA fiber with higher crystallinity. It has been found that temperature is essential and significant for an effective stereo-complexation process [15,33,34,35]. When the heat-setting temperature is lower than the melting point of HC, PLLA, and PDLA, segments do not have enough mobility to move to the vicinity of their enantiomers, so only a small amount of SC structures can be formed. When the heat-setting temperature is higher than the melting point of HC, in addition to SC spontaneous crystallization from the amorphous region, some of the HC were melted and rearranged to form new SC crystallites because the original SC crystallites acted as a template during the tension heat-setting process [35,36,37,38].

### 2.2. Thermal Properties

Figure 3 shows DSC heating curves of drawn−fiber, SC−160, SC−170, SC−180, and SC−190. Both drawn−fiber and SC−PLA fiber had an obvious melting endothermic peak near 222.5 °C, corresponding to the melting peak of SC crystallites. However, PLLA/PDLA drawn−fiber and SC−160 also had a weak melting endothermic peak of HC at about 168.0 °C, meaning that HC crystallites and SC crystallites coexisted in the fibers. As the tension heat-setting temperature increased, the melting peak of HC crystallites weakened. When the tension heat-setting temperature increased to 180 °C or 190 °C, the melting peak of HC crystallites disappeared and only a melting peak of SC crystallites existed in the DSC heating curve. A small melting peak of HC crystallites and a large melting peak of SC crystallites were observed in the DSC curve of drawn−fiber, which contradicts the WAXD profile, which lacks a SC crystalline peak (Figure 1a). This seeming contradiction can be explained by the melting of HC crystallites and transformation into SC crystallites during DSC heating [39]. In this study, PLA fiber with complete SC structure was obtained using a lower heat treatment temperature on a polyester fiber production line without additives compared with previous researches [40]. It provides a simpler route to produce PLA fiber with a high melting temperature.

The thermal parameters of PLLA/PDLA fiber are listed in Table 2. Since the DSC curve of the drawn−fiber had a cold crystallization peak (including HC cold crystallization and SC cold crystallization) at around 80 °C, and the ratio of HC cold crystallization and SC cold crystallization cannot be distinguished by DSC, Equations (7) and (8) (Section 3.4.2) cannot be used to accurately calculate the crystallinity and stereo-complex rate of drawn−fiber. It can be seen that, with the increase in the tension heat-setting temperature, the stereo-complex rate increased significantly, which was in accordance with the WAXD results. When the tension heat-setting temperature increased to 190 °C, the stereo-complex rate in PLLA/PDLA fiber could reach almost 100%, indicating that high-temperature tension heat-setting can effectively promote the formation of SC crystallites of PLLA/PDLA fiber. The crystallinity and stereo-complex rate determined by DSC tests were higher than those of WAXD, because the presence of SC crystallites seeded the crystallization of the amorphous region during the DSC heating process.

### 2.3. Thermal Stability

Figure 4 shows the thermal weight loss curves of the drawn−fiber and SC−PLA fiber. The temperature at 5% mass loss (T_5%_), the maximum decomposition temperature (T_max_), and the decomposition end temperature (T_end_) of PLLA/PDLA fiber are listed in Table 3. As illustrated in Figure 4 and Table 3, as the tension heat-setting temperature increased, the thermal stability of PLLA/PDLA fiber showed a downward trend. SC−190 fiber had the worst thermal stability, although the DSC results demonstrated its SC crystallites’ ratio was as high as 100%, and T_5%_ was lower than that of drawn−fiber by nearly 30 °C. This may be due to the occurrence of thermal degradation of PLA at high heat-setting temperatures, which caused the decrease in thermal stability of the fiber. 

### 2.4. Molecular Weight and Its Distribution

From the TGA tests describe above, it is known that the thermal stability of SC−PLA fiber decreased, which may result from the decrease in molecular weight, therefore, GPC was adopted to test the molecular weight of drawn−fiber and SC−PLA fiber. Table 4 shows the number average molecular weight (M_n_), weight average molecular weight (M_w_), and molecular weight distribution (PDI, M_w_/M_n_) of the drawn−fiber and SC−PLA fiber. Notably, both the drawn−fiber and SC−PLA fiber are PLLA/PDLA racemic blends, so their molecular weights are between the molecular weights of PLLA (M_w_ = 17.3 × 10^4^ g/mol) and PDLA (M_w_ = 4.5 × 10^4^ g/mol), and the PDI is significantly higher than that of PLLA and PDLA. Compared with drawn−fiber, the molecular weight of SC−PLA fiber after high-temperature tension heat-setting treatment decreased to different degrees. When the tension heat-setting temperature reached 190 °C, the M_n_ of SC−190 dropped to 12,667, the M_w_ dropped to 69,412, and the PDI increased to 5.48. This may be ascribed to the thermal degradation of PLA during the high-temperature tension heat-setting process and the random chain scission of the molecular chain.

### 2.5. Mechanical Properties

Figure 5 shows the monofilament strength of PLLA/PDLA drawn−fiber and SC−PLA fiber. The results showed that the monofilament strength of drawn−fiber was 2.80 cN/dtex (For PLA fiber, 1.0 cN/dtex is roughly equivalent to 128.0 MPa in numerical terms). Although the SC crystallites have better mechanical properties than the HC due to their compact molecular arrangement and strong intermolecular forces [12,41], the monofilament strength of SC−PLA fiber decreased compared with that of drawn−fiber. As the temperature of tension heat-setting increased, the monofilament strength of the fiber decreased. When the tension heat-setting temperature was up to 190 °C, the SC−fiber had a low monofilament strength of 1.95 cN/detx. This is similar to the results of Furuhashi et al. [31,42], who ascribed the decrease in mechanical properties to the decrease in orientation degree ($f_{C}$) of the crystal regions (Table 1). In addition, combined with the molecular weight test results (Table 4), the decrease in mechanical properties is also related to the decrease in molecular weight caused by thermal degradation of PLA during the tension heat-setting process.

### 2.6. Hydrolysis Resistance

#### 2.6.1. Changes in Crystallization Performance

The hydrolysis resistance of PLLA/PDLA fiber was investigated by accelerated hydrolysis tests in a 95 °C water bath. Figure 6 shows the WAXD patterns of drawn−fiber and SC−190 fiber exposed to accelerated hydrolysis for different times. Table 5 lists the crystallization parameters of PLLA/PDLA fiber versus hydrolysis time. The drawn−fiber presented a clear WAXD pattern dominated by amorphous regions, and the characteristic diffraction of HC at 2θ ≈ 16.4° was observed. After accelerated hydrolysis, the diffraction peaks of SC crystallites appeared at 2θ ≈ 11.9°, 2θ ≈ 20.6°, 2θ ≈ 23.8°. With prolonged hydrolysis time, the diffraction peak intensity of both the SC crystallites and HC crystallites increased, while the crystallinity and the absolute content of SC crystallites for the drawn−fiber increased. This implies that the amorphous region, HC crystallites and SC crystallites of the drawn−fiber were hydrolyzed to different degrees. Furthermore, the amorphous region had the fastest hydrolysis rate, followed by HC, while the SC crystallites had the slowest hydrolysis rate due to their stronger intermolecular forces and more stable crystal structure [12,16,32,43]. Therefore, the amorphous region no longer dominated, and the diffraction peaks of SC crystallites increased significantly for the drawn−fiber after accelerated hydrolysis. As seen from the WAXD patterns of SC−190 samples during the accelerated hydrolysis, the characteristic diffraction for SC crystallites were predominantly observed, while a weak diffraction peak corresponding to HC appeared near 2θ ≈ 16.7° with the extension of the hydrolysis time. This may be ascribed to the crystallization of partially amorphous regions to form HC at 95 °C in the water bath. The changes of SC−190 in crystallization performance after hydrolysis was similar to that of drawn−fiber, and the crystallinity and absolute content of SC crystallites increased significantly.

#### 2.6.2. Changes in Molecular Weight and Distribution

Table 6 shows the M_n_, M_w_, and PDI of drawn−fiber and SC−190 fiber hydrolyzed for different times. It is found that the molecular weight of drawn−fiber and SC−190 fiber decreased significantly after accelerated hydrolysis at 95 °C. When the hydrolysis time reached 32 h, the M_n_ and M_w_ of drawn−fiber dropped to 14.2% and 6.1% of the original, respectively, while the M_n_ for SC−190 fiber dropped to 31.0% and the M_w_ dropped to 13.3% of the original. The results showed that SC−190 fiber had a higher molecular weight retention, indicating better hydrolysis resistance.

#### 2.6.3. Changes in Thermal Stability

To reveal the changes in thermal stability of hydrolyzed fiber, TGA measurements were performed. TGA curves of the fiber samples with different hydrolysis times are shown in Figure 7. And the T_5%_, T_max_, and T_end_ of drawn−fiber and SC−190 fiber hydrolyzed for different times are listed in Table 7. The T_5%_ of the unhydrolyzed drawn−fiber was 307.1 °C, but this dropped to 270.4 °C after hydrolysis for 32 h. The T_max_ and T_end_ of drawn−fiber also exhibited a similar downward trend. This may be attributed to the pronounced decrease in molecular weight of drawn−fiber in the process of hydrolysis (Table 6), resulting in deterioration of its thermal stability. The T_5%_ of SC−190 also decreased gradually with the increase in hydrolysis time. However, its T_max_ and T_end_ were significantly different from that of drawn−fiber, which displayed an increasing trend after hydrolysis. The T_max_ of unhydrolyzed SC−190 was 333.8 °C, which rose to 346.2 °C after hydrolysis for 32 h. Although the absolute content of SC crystallites of drawn−fiber increased after hydrolysis, the T_max_ decreased significantly due to poor hydrolysis resistance and the dramatic decline in molecular weight after hydrolysis. SC−190 fiber with almost complete SC structure exhibited better hydrolysis resistance and therefore less molecular weight drop. In addition, SC crystallites provides better thermal stability because of strong intermolecular forces. The increase in the absolute content of SC is beneficial to the improvement of thermal stability of SC−PLA fiber. After hydrolysis, the improvement of thermal stability caused by the increase in SC absolute content was greater than the decrease caused by the decrease in molecular weight, and, as a consequence, the thermal stability of SC−190 fiber was increased.

#### 2.6.4. Changes in Mechanical Properties

To confirm the effects of formation of SC crystallites on the hydrolysis resistance of fiber, the monofilament strength retention rate was determined, and the results are summarized in Figure 8. With the extension of hydrolysis time, the monofilament strength of drawn−fiber and SC−PLA fiber showed a decreasing trend. The monofilament strength retention rate of SC−190 fiber hydrolyzed for 24 h had the highest value of 78.5%. In contrast, the monofilament strength of drawn−fiber, SC−160, and SC−170 after 12 h hydrolysis dropped to an untestable level. It is suggested that the mechanical properties of SC−190 fiber were relatively stable, i.e., its hydrolysis resistance was better. This is ascribed to the high crystallinity and complete SC structure of SC−190 fiber as well as the strong intermolecular force in the SC crystallites. Therefore, water molecules could not easily diffuse into the crystal region where the segments are closely arranged, resulting in better hydrolysis resistance, which can expand the application prospect of PLA fiber.

#### 2.6.5. Changes in Fiber Morphology

Figure 9 shows SEM images of drawn−fiber and SC−190. From Figure 9A,B, the diameters of drawn−fiber and SC−190 were similar, at about 17~18 μm. The surface of drawn−fiber was relatively smooth, while that of the SC−190 had a slight deformation due to the excessive heat-setting temperature. After hydrolysis for 8 h, both drawn−fiber and SC−190 can maintain their fiber morphology (Figure 9C,D). However, the drawn−fibers were obviously broken and could not maintain an intact fiber morphology after hydrolysis for 32 h (Figure 9E). In contrast, SC−190 could still maintain perfect fiber morphology (Figure 9F) and had certain mechanical properties (Figure 8). This is due to the better hydrolysis resistance of the SC−190 internal SC crystallites, which effectively improved the hydrolysis resistance of SC−190.

## 3. Materials and Methods

### 3.1. Materials

Commercially available PLLA (trade mark Luminy^®^ L130, with M_w_ = 17.3 × 10^4^ g/mol, PDI = 1.95, and optical purity ≥ 99%) was supplied by Total Corbion (Amsterdam, The Netherlands). PDLA with M_w_ = 4.5 × 10^4^ g/mol and PDI = 1.96 was purchased from Heilongjiang Xinda Enterprise Group Co., Ltd. (Harbin, China). The content of D-lactide in the PDLA is approximately 99%. 

### 3.2. Sample Preparation and Stereo-Complexation

The production process, including pre-spinning and post-spinning, was performed on a polyester fiber production line. The pre-spinning process included drying slices, blended melt-spinning and winding shaping. PLLA and PDLA pellets (50:50, wt%) were dried in a vacuum drum. The drying process was as follows: feeding at drum temperature < 60 °C, cooling for 0.5 h after feeding, and then drying at 60 °C for 1 h, 70 °C for 4 h, 80 °C for 3 h, and 120 °C for 6.5 h. The total drying period was 15 h, and the vacuum was ≤−0.098 MPa. The pre-spinning temperatures are listed in Table 8, and the spinning speed was 1020 m/min; metering pump frequency was 29.77 Hz. PLLA/PDLA as-spun fiber was obtained after pre-spinning.

The post-spinning process of the PLLA/PDLA fiber is shown in Figure 10. The drawing process consisted of two steps: water bath drawing at 62 ± 2 °C and steam drawing. The total drawing ratio was up to 3 times. To obtain PLLA/PDLA fiber with different stereo-complex rate, a high-temperature tension heat-setting process at 160 °C, 170 °C, 180 °C, and 190 °C was adopted respectively.

The PLLA/PDLA fiber after the drawing process was named drawn−fiber. The drawn−fiber after high-temperature tension heat-setting treatment was named SC−PLA fiber, and the fiber corresponding to 160 °C, 170 °C, 180 °C, and 190 °C tension heat-setting treatment was named SC−160, SC−170, SC−180, and SC−190, respectively.

### 3.3. Accelerated Hydrolysis Tests

Since textile dyeing and printing processes are often performed in a high-temperature and high-humidity environment, accelerated hydrolysis at 95 °C was used to characterize the hydrolysis resistance of drawn−fiber and SC−PLA fiber. The fiber samples including drawn−fiber, SC−160, SC−170, SC−180, and SC−190 were put into a 95 °C constant-temperature water bath for different times (4 h, 8 h, 12 h, 16 h, 24 h, or 32 h), and then rinsed with distilled water. The residual water in the hydrolyzed fiber samples was absorbed by long-fiber paper towels, and then the samples were dried at room temperature and transferred to a desiccator. Before the characterization, all fiber samples were placed in a standard test environment (65% relative humidity, 21 °C) for 24 h.

### 3.4. Accelerated Hydrolysis Tests

#### 3.4.1. Wide-angle X-ray Diffraction (WAXD)

The crystal structure was characterized using an X-ray diffractometer (Bruker D8 Advance, Bremen, Germany) equipped with a Ni-filtered Cu Kα radiation source (λ = 0.154 nm). The measurements were carried out at 40 kV and 150 mA with scan angles from 5° to 60° at a scan rate of 2°/min.

The obtained patterns were split and then the crystallinity and other related parameters of the fiber samples were calculated. The percentages of the diffraction peak areas of HC crystallites and SC crystallites to the total area of the diffraction pattern were used to represent the HC crystallinity ($X_{C,H}$, the absolute content of HC crystallites) and the SC crystallites crystallinity ($X_{C,S}$, the absolute content of SC crystallites) respectively. $X_{c}$ represents the total crystallinity of the fiber samples. $f_{\mathrm{SC}}$ represents the stereo-complex rate, which is the percentage of SC crystallites in the total crystals. The specific calculation formulas were as follows:(1)$$X_{C,H}=\frac{S_{\mathrm{hc}}}{S_{\mathrm{sc}}+S_{\mathrm{hc}}+S_{a}}\times 100\%$$
(2)$$X_{C,S}=\frac{S_{\mathrm{sc}}}{S_{\mathrm{sc}}+S_{\mathrm{hc}}+S_{a}}\times 100\%$$
(3)$$X_{c}=X_{c,S}+X_{c,H}$$
(4)$$f_{\mathrm{SC}}=\frac{X_{c,S}}{X_{c,S}+X_{c,H}}\times 100\%$$
where S_sc_, S_hc_, and S_a_ are the integral area of the SC crystallites, HC and the amorphous region, respectively. 

The grain size was calculated via the Scherer formula:(5)$$D=\frac{K\gamma }{\mathrm{Bcos}\theta }$$
where D is the grain size (Å); K is the Scherrer constant, about 0.89; B is the half-height width of the diffraction peak of the measured sample; θ is the diffraction angle (°); and γ is the X-ray wavelength, about 0.154 nm.

The orientation structure of the crystal region was tested using a D/max-2550VB/PC (RigaKu, Tokyo, Japan). The measurements were carried out at 40 kV and 150 mA with scan angles from −90° to 270° and a scan rate of 5°/min. The orientation factor ($f_{C}$) of the crystal region was calculated by the following formula.
(6)$$f_{C}=\frac{360{}^{\circ}-\sum H_{i}}{360{}^{\circ}}\times 100\%$$
where $H_{i}$ is the ith half-width of the peak in the azimuthal scan from WAXD determination.

#### 3.4.2. Differential Scanning Calorimetry (DSC)

DSC measurements of fiber samples were performed on a DSC8500 (PerkinElmer Inc., Waltham, MA, USA) under a nitrogen atmosphere at a flow rate of 50 mL/min. The temperature range was between 50 °C and 250 °C and the heating rate was set at 10 °C/min.

The ratio of the sum of melting enthalpy of HC crystallites and SC crystallites to the melting enthalpy of complete crystallization was used to express the degree of crystallinity (Xc). The melting enthalpy of SC crystallites as a percentage of the melting enthalpy of HC crystallites and SC crystallites is expressed as the content of SC crystallites ($f_{\mathrm{SC}}$). Since the enthalpy of complete crystallization of HC crystallites and SC crystallites are different, following formulas were established to calculate Xc and $f_{\mathrm{SC}}$.
(7)$$\mathrm{Xc}=\frac{{\Delta H}_{m-H}}{{\Delta H}^{o}{}_{m-H}}+\frac{{\Delta H}_{m-S}}{{\Delta H}^{o}{}_{m-S}}\times 100\%$$
(8)$$f_{\mathrm{SC}}=\frac{\frac{{\Delta H}_{m-S}}{{\Delta H}^{o}{}_{m-S}}}{\frac{{\Delta H}_{m-H}}{{\Delta H}^{o}{}_{m-H}}+\frac{{\Delta H}_{m-S}}{{\Delta H}^{o}{}_{m-S}}}\times 100\%$$
where ΔH_m−H_ and ΔH_m−S_ are the melting enthalpy of HC crystallites and SC crystallites, respectively, ΔH^o^_m−H_ is the melting enthalpy of complete crystallization of HC, which is 93.7 J/g, and ΔH^o^_m−S_ is the melting enthalpy of complete crystallization of SC crystallites, which is 142.0 J/g [12].

#### 3.4.3. Thermogravimetric Analysis (TGA)

TGA (TG 209 F1, NETZSCH, Selb, Germany) was used for thermal stability tests. 5–8 mg of each sample was heated from 50 °C to 600 °C with a 10 °C/min heating rate under a nitrogen atmosphere.

#### 3.4.4. Mechanical Performance Test

A fiber tensile tester (XQ-1C, Shanghai New Fiber Instrument Co., Ltd., Shanghai, China) was employed to test the monofilament strength of the fiber. Before the test, a 200 cN weight was used for calibration. The test conditions were as follows: pre-tension was 0.25 cN, clamping distance was 10 mm, and extension rate was 10 mm/min. Each group of fiber samples was measured 30 times.

#### 3.4.5. Gel Permeation Chromatography (GPC) 

GPC data of the samples, including the weight-average (M_w_), number-average (M_n_) molecular weights, and molecular weight distribution (PDI, M_w_/M_n_), were obtained by gel permeation chromatography (PL-GPC50, Polymer Laboratories, Church Stretton, UK). Hexafluoroisopropanol was used as the mobile phase, the flow rate was 0.3 mL/min, and polystyrene was used as the standard.

#### 3.4.6. Morphological Observation 

The morphology of the fiber was observed by a scanning electron microscope (SEM) (Quanta 250, FEI Company, Brno, Czech Republic) at 5 kV. And coated with a thin layer of gold before examination.

## 4. Conclusions

In this work, PLLA/PDLA fiber with different stereo-complex rates were prepared using a polyester fiber production line. The effects of tension heat-setting temperature on the structure and properties of PLLA/PDLA fiber were systematically investigated. As the tension heat-setting temperature increased, the content of HC crystallites gradually decreased and even disappeared, while the content of SC crystallites increased. When the tension heat-setting temperature reached 190 °C, the fiber achieved almost complete SC structure, which had a melting point of 222.5 °C. However, the monofilament strength of this fiber was relatively low, only 1.95 cN/dtex. The hydrolysis resistance of PLLA/PDLA fiber samples was studied using an accelerated hydrolysis experiment at 95 °C in a water bath. The drawn−fiber was found to exhibit a significant increase in crystallinity and the proportion of SC crystallites with the extension of the hydrolysis time. After 24 h of accelerated hydrolysis, the monofilament strength retention rate of SC−190 fiber was 78.5%, indicating better hydrolysis resistance, while the monofilament strength of other PLLA/PDLA fiber was lost to the point that it could not be tested. This study highlights the use of a polyester fiber production line to produce complete stereo-complexation PLA fiber by blended melt-spinning of PLLA and PDLA in equal proportion. Herein, the tension heat-setting at an appropriate temperature is believed to be the key to the complete stereo-complexation in the PLA fiber with favorable heat resistance and hydrolysis resistance. However, it is necessary to balance the relationship between heat resistance, hydrolysis resistance, and mechanical properties to find suitable application areas (such as filling fiber fields).

## Figures and Tables

**Figure 1 molecules-27-07654-f001:**
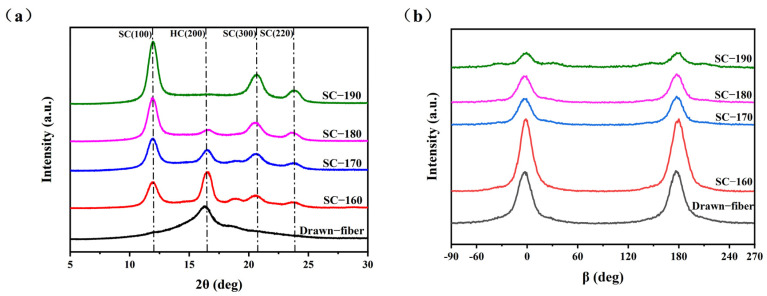
(**a**) WAXD patterns of crystal structure and (**b**) WAXD patterns of crystal orientation.

**Figure 2 molecules-27-07654-f002:**
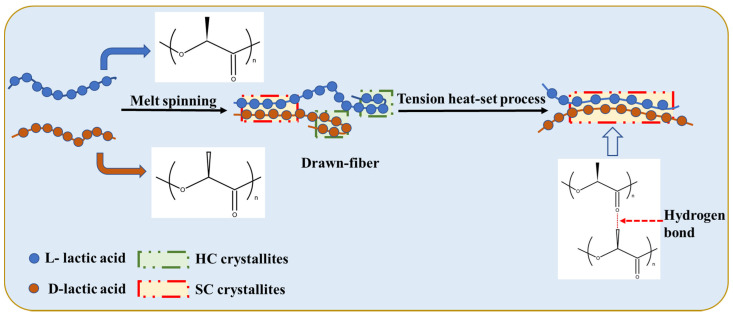
Schematic of stereo-complexation process for PLLA/PDLA fiber.

**Figure 3 molecules-27-07654-f003:**
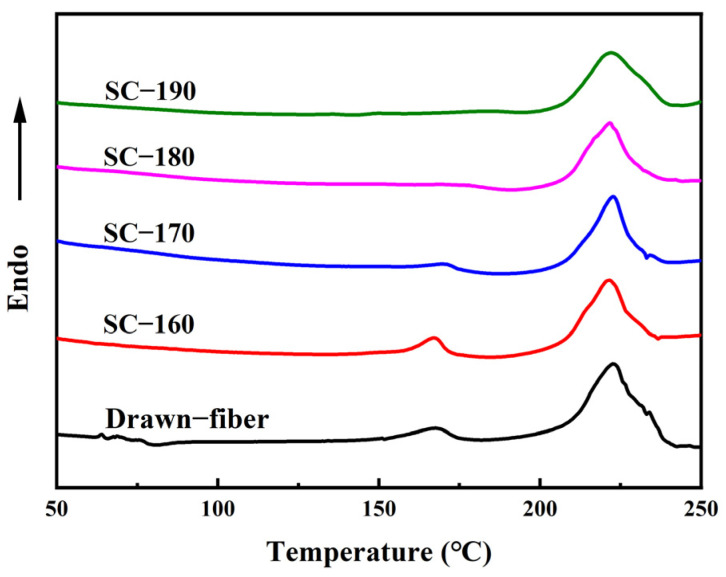
DSC heating curves of PLLA/PDLA fiber.

**Figure 4 molecules-27-07654-f004:**
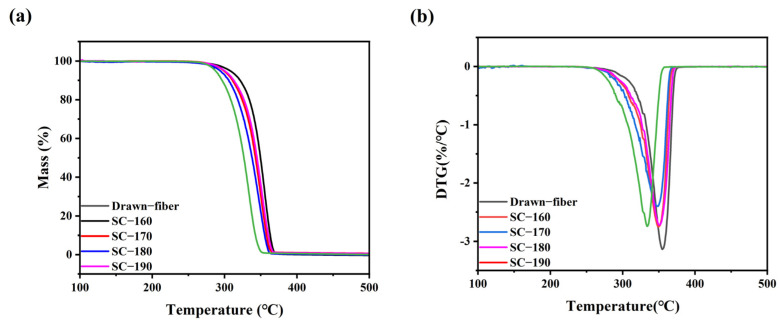
(**a**) TGA and (**b**) DTG curves of drawn−fiber and SC−PLA fiber.

**Figure 5 molecules-27-07654-f005:**
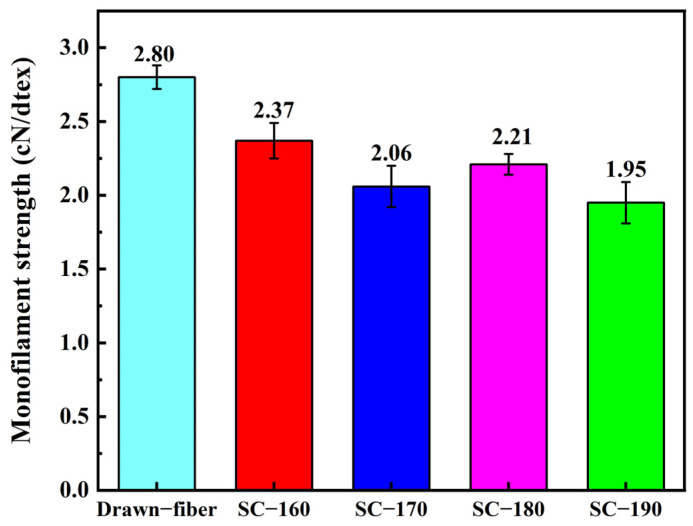
Monofilament strength of drawn−fiber and SC−PLA fiber.

**Figure 6 molecules-27-07654-f006:**
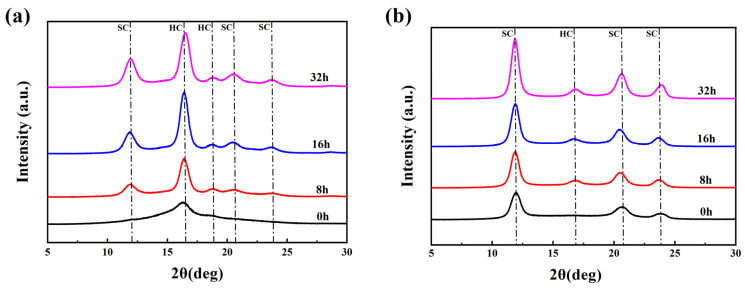
WAXD patterns of PLLA/PDLA fiber hydrolyzed for different times, (**a**) drawn−fiber, (**b**) SC−190.

**Figure 7 molecules-27-07654-f007:**
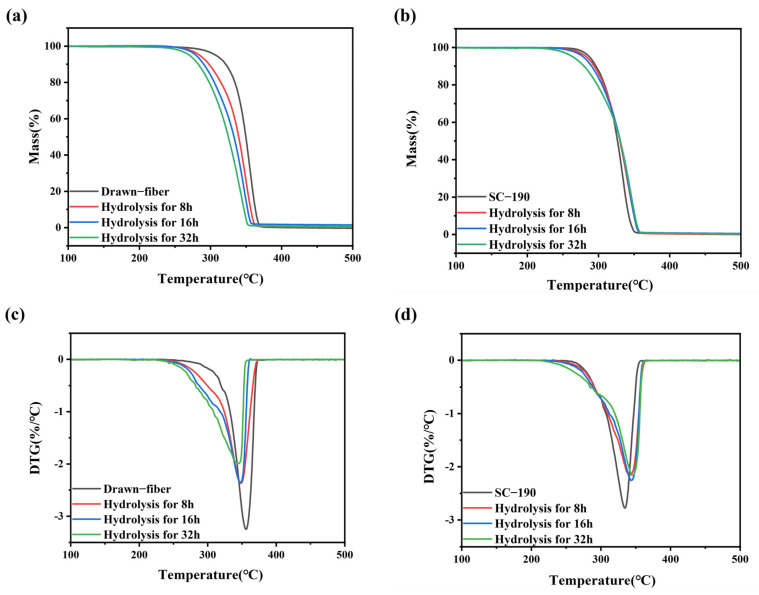
(**a**) TGA and (**c**) DTG curves of drawn−fiber and (**b**) TGA and (**d**) DTG curves of SC−190 fiber hydrolyzed for different times.

**Figure 8 molecules-27-07654-f008:**
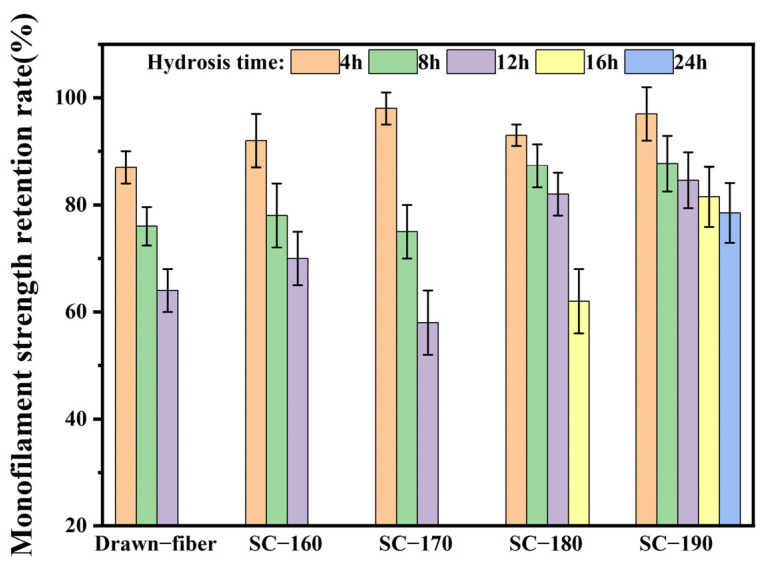
Monofilament strength retention rate of drawn−fiber and SC−PLA fiber hydrolyzed for different times.

**Figure 9 molecules-27-07654-f009:**
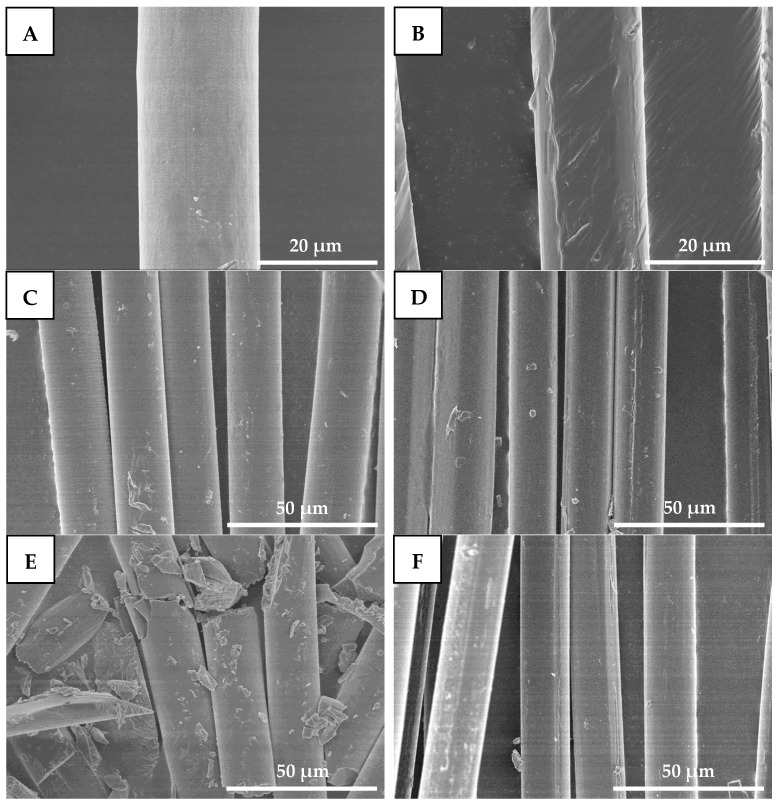
SEM micrographs of drawn−fiber hydrolyzed for different times ((**A**) 0 h, (**C**) 8 h, and (**E**) 32 h) and SC−190 hydrolyzed for different times ((**B**) 0 h, (**D**) 8 h, and (**F**) 32 h), (**A**,**B**) 2000×, (**C**–**F**) 1000×.

**Figure 10 molecules-27-07654-f010:**
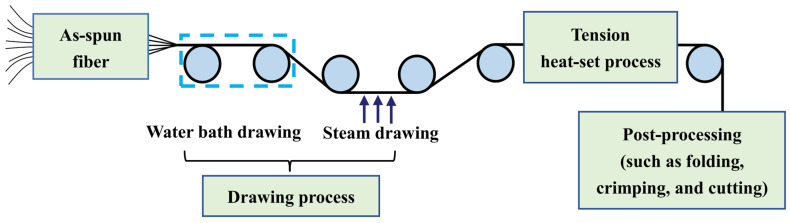
Flow chart of post-spinning process of PLLA/PDLA fiber.

**Table 1 molecules-27-07654-t001:** Crystallization parameters of drawn−fiber and SC−PLA fiber.

Samples	X_c_ (%)	f_SC_ (%)	Grain Size (Crystal Face (100), Å)	$f_{C}$ (%)
Drawn−fiber	28.8	14.3	-	84.9
SC−160	53.0	62.2	79.0	86.7
SC−170	45.3	81.4	82.0	86.2
SC−180	54.7	93.0	86.0	86.2
SC−190	58.5	98.7	90.0	78.7

**Table 2 molecules-27-07654-t002:** Thermal parameters of drawn−fiber and SC−PLA fiber.

Samples	ΔH_m−H_ (J/g)	ΔH_m−S_ (J/g)	X_c_ (%)	$f_{C}$ (%)
Drawn−fiber	11.4	74.7	-	-
SC−160	10.0	63.8	55.6	80.8
SC−170	3.4	64.4	49.0	92.6
SC−180	2.2	67.0	49.5	95.3
SC−190	0.0	74.5	52.5	100.0

**Table 3 molecules-27-07654-t003:** Thermal decomposition temperature of drawn−fiber and SC−PLA fiber.

	Drawn−fiber	SC−160	SC−170	SC−180	SC−190
T_5%_ (°C)	307.1	300.3	293.4	301.2	285.9
T_max_ (°C)	355.9	350.2	348.4	350.7	333.8
T_end_ (°C)	375.4	367.7	367.0	371.2	356.4

**Table 4 molecules-27-07654-t004:** Molecular weight and distribution of drawn−fiber and SC−PLA fiber.

Samples	M_n_	M_w_	PDI
Drawn−fiber	20,684	87,702	4.24
SC−160	15,289	73,070	4.78
SC−170	15,011	64,739	4.32
SC−180	15,302	70,046	4.58
SC−190	12,667	69,412	5.48

**Table 5 molecules-27-07654-t005:** Crystallization parameters of PLLA/PDLA fiber hydrolyzed for different times.

Sample	Hydrolysis Time (h)	$X_{C,H}$ (%)	$X_{C,S}$ (%)	X_c_ (%)	$f_{\mathrm{SC}}$ (%)
Drawn−fiber	0	24.7	4.1	28.8	14.3
	8	35.2	25.7	60.9	42.2
	16	33.6	30.8	64.4	47.9
	32	32.1	38.4	70.5	54.4
SC−190	0	0.8	57.7	58.5	98.7
	8	4.1	66.1	70.2	94.1
	16	12.0	68.7	80.7	85.1
	32	11.8	72.3	84.1	86.4

**Table 6 molecules-27-07654-t006:** Molecular weight and distribution of PLLA/PDLA fiber hydrolyzed for different times.

Samples	Hydrolysis Time (h)	M_n_	M_w_	PDI
Drawn−fiber	0	20,684	87,702	4.24
	8	10,736	28,372	2.64
	16	6088	19,425	3.19
	32	2939	5332	1.81
SC−190	0	12,667	69,412	5.48
	8	8690	23,925	2.75
	16	5540	10,908	1.97
	32	3931	9197	2.34

**Table 7 molecules-27-07654-t007:** Thermal decomposition temperature of drawn−fiber and SC−190 fiber hydrolyzed for different times.

Samples	Hydrolysis Time (h)	T_5%_ (°C)	T_max_ (°C)	T_end_ (°C)
Drawn−fiber	0	307.1	355.9	375.4
	8	284.0	348.4	361.2
	16	280.2	347.0	357.3
	32	270.4	343.2	352.4
SC−190	0	285.9	333.8	356.4
	8	281.9	342.4	363.0
	16	278.4	344.3	361.7
	32	263.4	346.2	363.4

**Table 8 molecules-27-07654-t008:** Pre-spinning temperature parameters.

Temperature of Each Zone of screw Extruder (°C)							Spinning Filter Temperature (°C)	Spinning Beam Temperature (°C)
Cooling Zone	I	II	III	IV	V	VI		
<100	220 ± 3	230 ± 3	235 ± 3	240 ± 3	235 ± 3	235 ± 3	235 ± 3	235 ± 3

## Data Availability

Not applicable.

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
