# Peer review of "Scalable Preparation of Complete Stereo-Complexation Polylactic Acid Fiber and Its Hydrolysis Resistance"

_molecules, 2022, doi:10.3390/molecules27217654_

Round 1

Reviewer 1 Report

In the present document the authors present a work based on the evaluation of the effect of post processing of PLLA-PDLA blends on the crystallization process, increasing it by stereo-complexation.

This is a very interesting work, although as the authors have reported, long time studied (ref. 39, 2003). They have focused their study in the influence of this SC crystallization of the hydrolytic character of the blends.

The document is very well written, and the results are concisely described and discussed. Minor details can be reviewed to improve the scientific character.

-       Pg 6. The authors report values of mechanical strength of the monofilaments in cN/dtex units. Even including this units, can they also add units in the IS (MPa)?

-       Pg 7. The authors report a hydrolytical study of samples of drawn-fibres and SC-190 at 95ºC. Why did they use this temperature? The hydrolytic process increases SC/HC crystallinity due to the molecules have some mobility at 95ºC. I had suggested performing this study also at a temperature closer to that Tg (60ºC) to reduce this mobility and visualize the hydrolysis without this influence.

-       Pg. 13. The authors give the value of melting enthalpy of PLA (100% of crystallinity) of 93.7 J/g. Please inform the original bibliographic reference. 

-       Pg. 14. Please inform about the standards used for GPC characterization.

Author Response

Response to Reviewer 1 Comments

Point 1: Pg 6. The authors report values of mechanical strength of the monofilaments in cN/dtex units. Even including this units, can they also add units in the IS (MPa)?

Response 1: We are sorry that we neglected the fact that cN/dtex is only used in the fiber field and most researchers are not familiar with it. As suggested by the reviewer, we have added a numerical conversion relationship between cN/dtex and MPa (For PLA fiber, 1.0 cN/dtex is roughly equivalent to 128.0 MPa in numerical terms) where cN/dtex first appears in the paper.

Point 2: Pg 7. The authors report a hydrolytical study of samples of drawn-fibres and SC-190 at 95ºC. Why did they use this temperature? The hydrolytic process increases SC/HC crystallinity due to the molecules have some mobility at 95ºC. I had suggested performing this study also at a temperature closer to that Tg (60ºC) to reduce this mobility and visualize the hydrolysis without this influence.

Response 2: As the reviewer point out, the hydrolysis at 95°C will lead to an increase in HC crystallites, and we were aware of this. There are several reasons why 95°C was chosen as the hydrolysis temperature, on the one hand, fiber often dyeing and printing at high-temperature and humid environment, so hydrolysis at 95°C was more suitable for the actual situation; In addition, the hydrolysis of PLA is highly temperature dependent. The time period required for hydrolysis at 60°C is too long and the hydrolysis effect is not obvious. Therefore, after detailed and careful consideration, we decided to choose 95°C as the hydrolysis temperature in this paper and added the corresponding instructions in section 3.3. Thank you very much for your question, which is of great significance for thesis quality improvement.

Point 3: Pg. 14. The authors give the value of melting enthalpy of PLA (100% of crystallinity) of 93.7 J/g. Please inform the original bibliographic reference.

Response 3: As reviewer suggested, we have added original bibliographic reference.

Point 4: Pg. 14. Please inform about the standards used for GPC characterization.

Response 4: We are very sorry that we neglected the standards used in GPC tests. Polystyrene as a standard sample in GPC tests. We have made changes based on the reviewers' comments.

We tried our best to improve the manuscript and made some changes in the manuscript. These changes will not influence the content and framework of the paper. We appreciate for Editors/Reviewers’ warm work earnestly, and hope that the correction will meet with approval.

Once again, thank you very much for your comments and suggestions.

Reviewer 2 Report

I consider that your paper tried to have a applied approach, but your own conclusions mention that there's a lot of work to be done in order to identify an specific application product. So, is only the beggining.

In my personal opinion I don't prefer the structure "results before the methods", but it is very well presented in the order you choosed. 

My only question is related to this phrase (lines 39-40) "However, the textile dyeing and printing process needs to be carried out in a high-temperature and humid environment", any of your formulations is close to this industrial requirement?

Author Response

Response to Reviewer 2 Comments

Point 1: My only question is related to this phrase (lines 39-40) "However, the textile dyeing and printing process needs to be carried out in a high-temperature and humid environment", any of your formulations is close to this industrial requirement?

Response 1: Thank you very much for your suggestion. Indeed, as reviewer pointed out, there is a lot of work needs to be accomplished subsequently. Regarding your question about any of our formulations is close to this industrial requirement or not in a high-temperature and humid environment. So far, our paper has demonstrated the effectiveness of SC in increasing the heat resistance and hydrolysis resistance of PLA, as well as the possibility of preparing complete SC structure PLA fiber on a large scale, but other relevant tests for industrial applications are carrying out. We believe that the complete stereo-complexation polylactic acid fiber with favorable heat resistance and hydrolysis resistance will be close to the requirements of the dyeing and printing process for PLA textile.

We tried our best to improve the manuscript and made some changes in the manuscript. These changes will not influence the content and framework of the paper. We appreciate for Editors/Reviewers’ warm work earnestly, and hope that the correction will meet with approval.

Once again, thank you very much for your comments and suggestions.

Reviewer 3 Report

In the paper, a simple process of complete stereo-complexation PLA fiber with favorable heat resistance and hydrolysis resistance was mentioned. From my point of view, the work is well-done and I would like to recommend it for publication after minor revisions.

Q1: The degree of crystallinity of SC-160~190℃ increases gradually, why does the crystallinity of 170℃ decrease.

Q2: In the SC-190 curve shown in Figure 3, what causes the exothermic peak around 140℃ ?

Q3:Why is the thermal stability of the same SC-190 sample higher than that of the unhydrolyzed sample under the condition of reduced molecular weight after hydrolysis?

Author Response

Response to Reviewer 3 Comments

Point 1: The degree of crystallinity of SC-160~190℃ increases gradually, why does the crystallinity of 170℃ decrease.

Response 1: This question is very relevant and significant to explain the HC-to-SC transformation during the tension heat-set process, but we did not reflect it in the paper. Therefore, thank you very much for raising the issue, which is of great importance for the quality improvement of the paper. After our in-depth and careful consideration, the decrease in crystallinity of SC-170 compared to SC-160 is due to the fact that part of the HC crystallites started to melt at 170°C, but there was not enough energy to recrystallize and transform into SC for molten HC, and amorphous structure formed during the cooling process. In contrast, SC-180 and SC-190 were more susceptible to HC-to-SC transformation at higher temperatures, and thus both crystallinity and stereo-complex rate were increased. Therefore, in summary, SC-170 has the lowest crystallinity among the SC-PLA fiber.

Point 2: In the SC-190 curve shown in Figure 3, what causes the exothermic peak around 140℃?

Response 2: After our careful study we believe that the peak around 140℃ in the SC-190 curve shown in Figure 3 is not an exothermic peak, but probably a fluctuation of the DSC curve due to the disturbance of the airflow. We are very sorry for this. We have re-performed the DSC test on SC-190 and added the revised data in the corresponding position in the paper.

Point 3: Why is the thermal stability of the same SC-190 sample higher than that of the unhydrolyzed sample under the condition of reduced molecular weight after hydrolysis?

Response 3: We apologize for our failure to explain clearly the reasons for the changes in SC-190 stability in our paper. For SC-190 fiber with almost complete SC structure, it exhibited better hydrolysis resistance and therefore less molecular weight drop. In addition, SC crystallites provides better thermal stability because of its strong intermolecular force. The increase of the absolute content of SC is beneficial to the improvement of thermal stability of SC-PLA fiber. After hydrolysis, the improvement of thermal stability caused by the increase of SC absolute content was greater than the decrease caused by the decrease of molecular weight, as a consequence the thermal stability of SC-190 fiber increased. As Reviewer suggested, we have reorganized this part according to the Reviewer’s suggestion.

We tried our best to improve the manuscript and made some changes in the manuscript. These changes will not influence the content and framework of the paper. We appreciate for Editors/Reviewers’ warm work earnestly, and hope that the correction will meet with approval.

Once again, thank you very much for your comments and suggestions.